Social exclusion; Migrants; Poverty; Depression

**Corresponding author:**
Maria Melchior,
E-mail: maria.melchior@inserm.fr

# COVID-19-related mental health difficulties among marginalised populations: A literature review

Chaka Camara[1], Pamela J. Surkan[1,2], Judith Van Der Waerden[1], Andrea Tortelli[3,4], Naomi Downes[1], Cécile Vuillermoz[1] and Maria Melchior[1,4]

[1]Sorbonne Université, INSERM, Institut Pierre Louis d'Epidémiologie et de Santé Publique, Paris, France; [2]Department of International Health, Johns Hopkins University Bloomberg School of Public Health, Baltimore, MD, USA; [3]Pôle GHU Psychiatrie Précarité, Groupe Hospitalier Universitaire Paris Psychiatrie & Neurosciences, Paris, France and [4]Health Department, French Collaborative Institute on Migration, Aubervilliers, France

## Abstract

The COVID-19 pandemic has had negative consequences on the mental health of the population, which has been documented. Marginalised groups that are at risk of poor mental health overall have been particularly impacted. The purpose of this review is to describe the mental health impact of the COVID-19 pandemic on marginalised group (i.e. persons who are socio-economically disadvantaged, migrants and members of ethno-racial minorities, experience homelessness) and identified interventions which could be well-suited to prevent and address mental health difficulties. We conducted a literature review of systematic reviews on mental health difficulties since the beginning of the COVID-19 epidemic and appropriate interventions among marginalised groups published from January 1, 2020 to May 2, 2022, using Google Scholar and PubMed (MEDLINE). Among 792 studies on mental health difficulties among members of marginalised groups identified by keywords, 17 studies met our eligibility criteria. Twelve systematic reviews examining mental health difficulties in one or several marginalised groups during the COVID-19 pandemic and five systematic reviews on interventions that can mitigate the mental health impact of the COVID-19 pandemic were retained in our literature review. The mental health of marginalised groups was severely affected during the COVID-19 pandemic. Most frequently reported mental health difficulties included symptoms of anxiety and depression. Additionally, there are interventions that appear effective and well-suited for marginalised populations, which should be disseminated on a large scale to mitigate the psychiatric burden in these groups and at the population level.

## Impact statement

Marginalised populations – including persons who are socio-economically disadvantaged, migrants and members of ethno-racial minorities, experience homelessness – have taken an important public health toll because of the COVID-19 epidemic. In particular, they seem to have especially high levels of symptoms of anxiety and depression, which deserve special attention from health professionals and public health decision makers. There are potentially effective interventions that can help target marginalised groups and should be made widely available to limit the mental health burden in this group.

## Social media summary

Marginalized groups are at higher risk of experiencing mental health difficulties in the aftermath of the COVID-19 pandemic, yet few interventions have been shown to be effective in these populations.

## Background

Since the end of 2019, the COVID-19 pandemic has had detrimental consequences worldwide in terms of morbidity and mortality, has weakened healthcare systems and has led to the implementation of preventive measures which were disruptive in terms of social and economic life including lockdowns, workplace and school closing as well as other restrictions in daily activities (WHO, 2020; OECD, 2021). These aspects of the COVID-19 pandemic have resulted in a significant deterioration of population mental health (Fiorillo et al., 2020; Santomauro et al., 2021).

This situation was not entirely unexpected. Previous epidemics such as H1N1, MERS, Sars-Cov-1 and Ebola had highlighted the risk of an increase in levels of anxiety, depression and post-traumatic stress disorder (PTSD) (Park et al., 2020; Zürcher et al., 2022), particularly among frontline healthcare workers (Chigwedere et al., 2021; Magnavita et al., 2021; Yuan et al., 2021). Yet, the COVID-19 pandemic had a much broader scope, and in addition to healthcare workers, women, young people as well as persons experiencing chronic mental and physical disorders have also found to be at increased risk of psychological distress, as well as of symptoms of anxiety and depression (Salari et al., 2020; Chigwedere et al., 2021; Panchal et al., 2022). Several authors have suggested that the COVID-19 pandemic should be considered as a new source of mental health trauma (Adhanom Ghebreyesus, 2020; Bridgland et al., 2021). Moreover, different marginalised groups experienced particularly high increases in rates of psychiatric disorders, including persons who experience socio-economic disadvantage, the homeless, migrants and members of ethno-racial minority groups. These groups have disproportionately suffered from the direct and indirect impacts of the COVID-19 pandemic, in terms of health as well as economic and social impact (Wachtler et al., 2020; Green et al., 2021). Moreover, inequalities in mental health exacerbated by the COVID-19 pandemic could cause further socio-economic disadvantage in the foreseeable future. Additionally, marginalised groups tend to experience difficulties in accessing standard healthcare (Wachtler et al., 2020), which calls for innovative interventions. Prior systematic reviews and meta-analyses of the mental health impact of COVID-19 have extensively summarised findings on levels of psychological distress in the general population and among healthcare workers (Zürcher et al., 2020; Chigwedere et al., 2021; Magnavita et al., 2021; Yuan et al., 2021); however, to date there was no overview of the situation among marginalised groups. There is also evidence that in-person as well as online psychosocial interventions can be effective in reducing levels of stress, anxiety, depression and insomnia exacerbated by the COVID-19 pandemic (Lekagul et al., 2022; Ye et al., 2022). The aim of this literature review is to describe current evidence regarding the levels of psychological distress and psychiatric disorders among persons belonging to marginalised groups in the context of the COVID-19 pandemic, as well as promising interventions well-suited for these populations. Given the large number of studies recently published in this area and in order to analyse the most rigorous evidence, we focused our review on systematic reviews and meta-analyses.

## Methods

Marginalised groups considered in this review are persons who experience socio-economic disadvantage (i.e. low income, low educational level, a low-grade occupation or unemployment), migrants and persons belonging to an ethno-racial minority group and persons who experience homelessness. Regarding interventions, we included all studies describing programmes which could mitigate the mental health impact of the COVID-19 pandemic among marginalised groups. We defined mental health outcomes as participants' overall mental health, as well as symptoms of anxiety, depression, stress and psychological distress, which are the most frequent psychological difficulties in the general population. Additionally, we considered all interventions aiming to help persons cope with psychological distress during the COVID-19 pandemic.

To identify articles examining rates of psychological distress as well as psychiatric disorders among marginalised groups in the context of the COVID-19 pandemic, we proceeded as follows. An electronic search was subsequently conducted in Google scholar and PubMed between April 8, 2022 and May 2, 2022 to identify systematic reviews and/or meta-analyses published after March 2020. Using Boolean combinations (AND, OR, NOT), the following research terms were used: 'COVID-19'; 'SARS-CoV-2'; 'coronavirus'; 'vulnerable group'; 'vulnerability'; 'group at high-risk'; 'marginalised'; 'inequality'; 'income loss'; 'low-income'; 'household income'; 'socioeconomic disadvantaged'; 'unemployment'; 'homeless'; 'people experiencing homelessness'; 'migrant'; 'born abroad'; 'ethnic minority'; 'minority'; 'ethnic group'; 'ethno-racial minority'; 'mental health'; 'mental health disorder'; 'depression'; 'anxiety'; 'psychological distress'; 'suicide'.

Due to the dearth of current reviews on mental health interventions specifically designed for/targeted at marginalised groups, we decided to broaden the scope of the search to all intervention studies aiming to reduce mental health difficulties during the COVID-19 pandemic. Specifically, the terms searched for in Google scholar and PubMed were 'digital intervention'; 'intervention'; 'prevention'; 'COVID-19'; 'SARS-CoV-2'; 'coronavirus'; 'mental health'; 'mental disorder'; 'depression'; 'anxiety'; 'psychological distress'; 'suicide'; 'decrease'; 'reduction'.

Studies included in our review had to fulfil the following criteria: (1) systematic review and/or meta-analysis; (2) focus on mental health outcomes or a mental health intervention; (3) inclusion of at least one of the targeted marginalised groups; (4) implementation during the period of the COVID-19 pandemic and (5) English language. The time range for the publication of the selected papers was January 1, 2020 to May 2, 2022.

### Data collection

Data extraction was performed in Excel. The data extraction form relative to systematic analyses on mental health risk of marginalised groups included the following information: (1) authors; (2) journal; (3) date of publication; (4) title; (5) period covered; (6) number of articles included in the review; (7) countries; (8) study design; (9) populations studied; (10) mental health outcomes and (11) main results. Whenever possible, we also extracted p-values.

The data extraction form relative to systematic reviews of interventions aiming to reduce the impact of the COVID-19 pandemic on mental health included the following information: (1) authors, (2) journal, (3) setting, (4) characteristics of the studied population, (5) type of intervention and (6) mental health outcomes.

### Results

Our search retrieved 792 studies through searching the identified databases. After removing duplicates, we screened the titles and abstracts of the remaining studies. After reading the full text of the remaining studies, 17 studies met our eligibility criteria.

### Characteristics of studies describing mental health risks among marginalised populations

Overall, we identified 12 systematic reviews examining mental health difficulties in one or several marginalised groups during the COVID-19 pandemic and 5 systematic reviews on interventions that can mitigate the mental health impact of the COVID-19

**Table 1.** Systematic reviews examining the mental health of marginalised groups in the context of the COVID-19 pandemic: 2020–2022

| Authors (year) | Journal | Year of publication | Title | Period covered | Number of studies included | Countries studied | Design | Study design included | Marginalised groups studied | Outcomes studied | Main results |
|---|---|---|---|---|---|---|---|---|---|---|---|
| *Socio-economically disadvantaged groups* | | | | | | | | | | | |
| Xiong et al. (2020) | *Journal of Affective Disorders* | 2021 | Impact of COVID-19 pandemic on mental health in the general population: A systematic review | | 19 | Different countries (China = 10; USA/Western Europe = 6; others = 2) | Systematic review | | Employment status | Depression (DBI II, SDS, CES-D), anxiety (BAI, GAD-7, SAS), DASS-21, HADS, PTSD | In general, mental health has deteriorated during the COVID 19 pandemic. The unemployed have elevated levels of depression and anxiety |
| Gibson et al. (2021) | *Canadian Psychology* | 2021 | The impact of inequality on mental health outcomes during the COVID-19 pandemic: A systematic review | | 117 | Different countries (China = 47; USA = 14) | Systematic review | Cross-sectional studies (*n* = 112); cohort studies (*n* = 4); case control studies (*n* = 1) | Income; Employment and occupation status; Migrant status and ethnicity | Worry/nervousness/ anxiety–depression– stress/PTSS/PTSD | Participants with low income, unemployed, low occupational grade, migrant or belonging to ethnic minority groups at high risk of worry/anxiety/ depression |
| Wang et al. (2020) | *PLoS One* | 2020 | Factors associated with psychological distress during the coronavirus disease 2019 (COVID19) pandemic on the predominantly general population: A systematic review and meta-analysis | 12/2019–15/07/2020 | 68 | Western Pacific Region (*n* = 41; China = 39; Japan = 1; Vietnam = 1); European region (*n* = 16; Italy = 6; UK, Spain, Turkey = 2; Slovenia, Albania, France, Ireland = 1) Region of Americas (*n* = 4; UU = 3 Colombia = 1) Eastern Mediterranean region (*n* = 4; Iran, Israel, Saudi Arabia = 1); South-East Asia Region (*n* = 2; India); Africa (*n* = 1, Tunisia) | Systematic review and a meta-analysis | Cross-sectional studies | SES | Anxiety, depression, distress, stress, post-traumatic stress, and insomnia | Participants with low SES have higher levels of mental health disorders |
| Rodríguez-Fernández et al. (2021) | *International Journal of Environmental Research and Public Health* | 2021 | Psychological effects of home confinement and social distancing derived from COVID-19 in the general population – A systematic review | 12/2019 onward | 26 | China (*n* = 6); Spain (*n* = 3); Germany (*n* = 2); UK (*n* = 2); Saudi Arabia (*n* = 1); Brazil (*n* = 1); India (*n* = 1); South Korea (*n* = 1); Pakistan (*n* = 1); Jordan (*n* = 1); Italy (*n* = 1); Vietnam (*n* = 1); Turkey (*n* = 1); Bangladesh (*n* = 1); USA (*n* = 1); multiple countries (*n* = 2) | Systematic review | Cross-sectional (*n* = 24) and longitudinal studies (*n* = 2) | Education, socio-economic disadvantage | Anxiety, depression, stress, PTSD | Participants with low education or who experience socio-economic disadvantage have high levels of anxiety, depression and stress |
| Filindassi et al. (2022) | *COVID* | 2022 | Impact of the COVID-19 first wave on psychological and psychosocial dimensions: A systematic review | 12/2019–06/2020 | 294 | 30 different countries | Systematic review | Not reported | Employment status, Income, Education, SES | Anxiety, depression, stress, other mental health outcomes, social support, coping | Participants with low income and low levels of education found to have highest levels of anxiety and depression |
| Leung et al. (2022) | *Transnational Psychiatry* | 2022 | Mental disorders following COVID-19 and other epidemics: A systematic review and meta-analysis | Until 09/12/2020 | 255 (15 in the meta-analysis) | 50 different countries (China *n* = 64) | Systematic review and a meta-analysis | Not reported | Employment status, Income, Education | Anxiety, depression, post-traumatic stress disorder, psychological distress, acute stress, suicidality | Low-income groups, persons with low education severely impacted in terms of mental health by the sanitary crisis |

*(Continued)*

**Table 1.** (*Continued*)

| Authors (year) | Journal | Year of publication | Title | Period covered | Number of studies included | Countries studied | Design | Study design included | Marginalised groups studied | Outcomes studied | Main results |
|---|---|---|---|---|---|---|---|---|---|---|---|
| *People experiencing homelessness* | | | | | | | | | | | |
| Corey et al. (2022) | *International Journal of Environmental Research and Public Health* | 2022 | A scoping review of the health impact of the COVID-19 pandemic on persons experiencing homelessness in North America and Europe | | 96 | US ($n = 51$); UK ($n = 9$); France ($n = 9$); Canada ($n = 6$); Spain ($n = 5$), Italy ($n = 4$); Germany ($n = 4$); Denmark ($n = 2$); Belgium ($n = 2$); Multiple ($n = 2$); Slovakia ($n = 1$); Ireland ($n = 1$) | Scoping review | Cross-sectional ($n = 30$); unspecified ($n = 7$); longitudinal ($n = 4$); mixed-methods ($n = 4$); pilot ($n = 4$); case study ($n = 4$); qualitative ($n = 3$); report ($n = 3$); case report ($n = 2$); case series ($n = 2$); retrospective ($n = 2$) | Persons experiencing homelessness | Mental health (anxiety, depression, loneliness, nervousness, suicidal thoughts, psychological distress) | Persons experiencing homelessness are at high risk of mental health difficulties: high rates of anxiety, depression, loneliness |
| Rajkumar (2020) | *Asian Journal of Psychiatry* | 2020 | COVID-19 and mental health: A review of the existing literature | Unreported | 28 | China ($n = 18$); Iran ($n = 2$); Canada ($n = 2$); Brazil ($n = 1$); Singapore ($n = 1$); India ($n = 1$); Japan ($n = 1$); no specified ($n = 2$) | Literature review | Cross-sectional studies ($n = 4$); letters/commentaries ($n = 24$) | Persons experiencing homelessness, migrant workers | Anxiety, depression, sleep disorders | The COVID-19 pandemic had a significant impact on mental health of persons who are homeless |
| Tsamakis et al. (2021) | *Experimental and Therapeutic Medicine* | 2021 | COVID-19 and its consequences on mental health | Unreported | Unreported | Unreported | Review | – | Persons experiencing homelessness, refugees | Mental health disorders (anxiety, depression, stress, PTSD) | Impact of the COVID-19 pandemic on mental health of persons who are homeless, but the evidence is limited |
| Uphoff et al. (2021) | *PLoS One* | 2021 | Mental health among healthcare workers and other vulnerable groups during the COVID-19 pandemic and other coronavirus outbreaks: A rapid systematic review | 07/2020–08/2020 | 25 | Different countries | Rapid systematic review | Empirical studies ($n = 10$); cross-sectional studies ($n = 4$); experimental or observational studies with a control group ($n = 1$); qualitative studies ($n = 1$) | Persons experiencing homelessness ($n = 1$) | Any mental health condition, quality of life, suicide or attempted suicide | Lack of evidence regarding the mental health of persons who are homeless |
| *Migrants, ethno-racial minorities* | | | | | | | | | | | |
| Gibson et al. (2021) | *Canadian Psychology* | 2021 | The impact of inequality on mental health outcomes during the COVID-19 pandemic: A systematic review | | 117 | Different countries ($n = 28$)/China = 47; USA = 14 | Systematic review | Cross-sectional studies ($n = 112$); cohort studies ($n = 4$); case control studies ($n = 1$) | Education; income; employment and occupation status; migrant status and ethnicity | Worry/nervousness/anxiety–depression–Stress/PTSS/PTSD | Migrant participants at risk of mental health difficulties. One study found no association; another study found worse mental health among persons who self-identified as White |
| Hintermeier et al. (2021) | *Journal of Migration and Health* | 2021 | SARS-CoV-2 among migrants and forcibly displaced populations: A rapid systematic review | Since 12/2019 | 15 | High income countries ($n = 7$); upper or low middle-income countries ($n = 5$); low-income countries ($n = 3$) | Rapid systematic review | | Refugees, asylum seekers ($n = 5$); migrant workers ($n = 4$); international students ($n = 2$); migrants with no further specification ($n = 2$) | Anxiety, depression (PHQ2; GAD2, PSS-4); social well-being, loneliness | Among migrants and displaced persons, negative impacts of the COBID-19 crisis |
| Jesline et al. (2021) | *Humanities and Social Sciences Communications* | 2021 | The plight of migrants during COVID-19 and the impact of circular migration in India: A systematic review | | 15 | India | Systematic review | | Migrant workers | Loneliness; anxiety; irritability; depression | One study reported a high prevalence of loneliness and depression among migrants |
| Tsamakis et al. (2021) | *Experimental and Therapeutic Medicine* | 2021 | COVID-19 and its consequences on mental health | Unreported | Unreported | Unreported | Review | – | Persons experiencing homelessness; refugees | Mental health disorders (anxiety, depression, stress, PTSD) | Limited information on migrants |

pandemic. Four systematic reviews also included a meta-analysis. The number of included studies per article ranged from 15 to 294 (Table 1).

The systematic reviews we identified mainly included cross-sectional studies and were mostly based in China or Western industrialised countries (USA, UK, France, Italy and Spain). The majority of systematic reviews presented data on the targeted marginalised groups as a subgroup of the population studied, except Hintermeier et al. (2021), Jesline et al. (2021) and Corey et al. (2022) who specifically focused on persons experiencing homelessness, migrants and displaced populations.

A majority of systematic reviews explored a diversity of mental health outcomes including depression (Xiong et al., 2020; Wang et al., 2020; Gibson et al., 2021; Hintermeier et al., 2021; Rodríguez-Fernández et al., 2021; Corey et al., 2022; Filindassi et al., 2022; Leung et al., 2022), anxiety (Xiong et al., 2020; Wang et al., 2020; Gibson et al., 2021; Hintermeier et al., 2021; Jesline et al., 2021; Rodríguez-Fernández et al., 2021; Corey et al., 2022; Filindassi et al., 2022; Leung et al., 2022), PTSD (Wang et al., 2020; Xiong et al., 2020; Gibson et al., 2021; Rodríguez-Fernández et al., 2021; Leung et al., 2022), stress (Wang et al., 2020; Xiong et al., 2020; Gibson et al., 2021; Rodríguez-Fernández et al., 2021; Filindassi et al., 2022; Leung et al., 2022), suicidal thoughts (Corey et al., 2022), self-harm, nervousness (Corey et al., 2022; Gibson et al., 2021) and well-being (Hintermeier et al., 2021; Filindassi et al., 2022). All included systematic reviews presented at least two different mental health outcomes.

Most included studies relied upon participants' self-reports of psychological distress or psychological difficulties, assessed using a variety of scales [e.g., the General Anxiety Disorder scale (GAD-7), Beck Anxiety Inventory (BAI), Patient Health Questionnaire (PHQ-9), Depression, Anxiety and Stress Scale (DASS), General Health Questionnaire (GHQ), Hospital Anxiety and Depression Scale (HADS), Perceived Stress Scale (PSS), Self-rating Depression Scale (SDS) and Self-rating Anxiety Scale (SAS)].

### Persons experiencing socio-economic disadvantage

Regarding rates of mental health difficulties and psychiatric disorders among persons experiencing socio-economic disadvantage, we observed the following. Wang et al. (2020) reported an elevated prevalence of anxiety [1.21 (1.05–1.40; $I^2 = 86.1\%$)], depression [1.15 (1.03–1.29; $I^2 = 82.0\%$)] and stress [1.15 (1.03–1.29); $I^2 = 9.0\%$] among persons with a low educational level, compared to those with an intermediate or high educational level across 30 studies. Rodríguez-Fernández et al. (2021) also found an increased prevalence of symptoms of anxiety (8.3–45.1% in five studies), depression (14.6–46.42% in seven studies), stress-related symptoms and PTSD (8.1–49.66% in four studies) among persons with a low level of education. Xiong et al. (2020) found that persons with a low level of education were more likely to suffer from anxiety (ranging from 6.33% to 50.9% in 11 studies) and depression (ranging from 14.6% to 48.3% in 12 studies), compared to persons with a high level of education. However, in this systematic review, a low level of education was not associated with symptoms of PTSD. Finally, Gibson et al. (2021), analysing 28 studies, found that a low level of education was associated with a deteriorated mental health. Conversely, the same systematic review also reported that in five studies a high level of education was associated with worse mental health outcomes and in four studies the level of education was not

associated with participants' mental health. Likewise, Filindassi et al. (2022) found a high rate of anxiety among highly educated groups in two studies. Nevertheless, stress and depression were significantly more frequent among persons with a low level of education in most studies included in this systematic review (Gibson et al., 2021; Rodríguez-Fernández et al., 2021; Wang et al., 2020; Xiong et al., 2020).

Similarly, six studies reported that individuals with a low income were also at increased risk of psychological distress during the COVID-19 pandemic. Wang et al. (2020) suggested that members of low-income groups were more likely to suffer from anxiety [1.45 (1.24–1.69; $I^2 = 82.3\%$)], depression [1.56 (1.26–1.92; $I^2 = 85.4\%$)] and stress [1.27 (1.20–1.34; $I^2 = 0\%$)] in comparison with members of higher-income groups. Gibson et al. (2021) and Rodríguez-Fernández et al. (2021) observed similar findings. Specifically, Rodríguez-Fernández et al. (2021) reported elevated levels of symptoms of anxiety (in eight studies), depression (in 10 studies) and PTSD (in six studies) among persons with low income. Additionally, Filindassi et al. (2022) found higher levels of symptoms of mental distress such as anxiety (in four studies), depression and stress (in three studies) among persons with low income. Xiong et al. (2020) showed a positive association between a low income and the risk of depression in two studies. Leung et al. (2022) found a pooled prevalence of 13.0% of psychological distress in the general population, with persons experiencing low income being at high risk. Persons belonging to a low-income group were also at increased risk of experiencing an acute stress disorder.

Across the systematic reviews analysed, there is consensus regarding the association between unemployment and mental health problems of the context of the COVID-19 epidemic. Xiong et al. (2020) reported that persons who were unemployed were at increased risk of developing depression (prevalence rates across 12 studies ranging from 14.6% to 48.3%) and stress symptoms (prevalence rates across four studies ranging from 8.1% to 81.9%). Leung et al. (2022) reported prevalence rates of anxiety ranging from 14% to 32.8% (in six studies) and of depression ranging from 9.5% to 27.8% (in 12 studies) with persons who were unemployed disproportionately affected by these symptoms. Rodríguez-Fernández et al. (2021) and Filindassi et al. (2022) also reported elevated levels of symptoms of anxiety, stress and psychological distress among persons who were unemployed. These results corroborate those of the meta-analysis performed by Wang et al. (2020): in this analysis persons who were employed had a pooled OR of psychological distress of 0.89 (0.78–1.02; $I^2 = 26.6\%$) in relation to those who were unemployed, highlighting a protective effect of employment. Gibson et al. (2021) confirmed this finding. Noticeably, two studies reported by Gibson et al. (2021) found that being on temporary leave because of the COVID-19 pandemic was more strongly associated with poor mental health than unemployment. The conclusion of Gibson et al. (2021) contrasts with findings of others studies and indicates heterogeneity in the risk of psychological distress according to employment status.

### Migrants and members of ethno-racial minority groups

Four systematic reviews examined levels of psychological distress among persons who are migrant or belong to an ethno-racial minority group, mostly suggesting an elevated risk of mental health

difficulties. Gibson et al. (2021) reported worse mental health among migrants and members of the Black, Asian and Minority Ethnic group in 10 studies. Hintermeier et al. (2021) examined five studies and reported that 73.5% of migrant workers felt anxiety, depression or perceived stress due to the COVID-19 pandemic, persons originating from South-East Asia being especially impacted. 63.3% of migrant workers reported an increase in negative thoughts, tension, frustration, irritability and fear of death. Two studies reported significantly elevated levels of distress among migrants (Hintermeier et al., 2021). Tsamakis et al. (2021) suggested that deterioration in mental health among migrants arises from disadvantaged living conditions, which also contribute to an increased risk of COVID-19 infection. Similarly, Jesline et al. (2021) also suggested that precarious living conditions in migrant populations contribute to high levels of psychological difficulties experienced in this group. This marginalised population, experiencing multiple forms of social vulnerability, has elevated odds of experiencing various negative outcomes: suicidal tendencies, self-harm, loneliness, anxiety and psychological distress.

However, two studies analysed by Gibson et al. (2021) did not report elevated rates of mental health difficulties among members of ethno-racial minority groups. Furthermore, at least one study conducted in the USA found that persons who identified as White Caucasian had higher levels of psychological disorders during the COVID-19 pandemic than those who identified as Asian or Hispanic. Although members of migrant and ethno-racial minority groups were overall at significantly higher risk of experiencing mental disorders, it is important to note that the extent to which this risk is elevated varies across different groups and settings.

### *People experiencing homelessness*

Corey et al. (2022) conducted an exhaustive literature review regarding issues affecting persons experiencing homelessness in the context of the COVID-19 pandemic. Five original studies revealed poor mental health in this group, with a 32% and 49% prevalence of symptoms of anxiety and of feelings of loneliness, respectively. Women experiencing homelessness and unstable housing appear particularly likely to have high levels of anxiety (42%) and depression (55%) according to nine studies. Moreover, Corey et al. (2022) demonstrated that the COVID-19 pandemic induced a deterioration in the mental health of persons experiencing homelessness. Additionally, two of the included studies indicated a deterioration of mental health (39% of persons) throughout the COVID-19 pandemic and a rise in thoughts of self-harm and suicide (up to 21%). To the contrary, three studies summarised in the review conducted by Corey et al. (2022), suggested improvements in mental health and well-being among persons experiencing homelessness during the COVID-19 pandemic. These studies conducted in Italy and Ireland were performed in night shelters that provided services 24 hours a day, 7 days a week, which could have mitigated the impact of the COVID-19 pandemic. Another study, analysed by Corey et al. (2022) and conducted among persons experiencing homelessness in France, indicated that 24% had unmet mental health needs. Several barriers to care were pointed out: barriers in access as well as insufficient efficacy of telephone and online services.

Three additional reviews indicated high levels of psychological distress among persons experiencing homelessness (Rajkumar, 2020; Tsamakis et al., 2021; Uphoff et al., 2021).

### *Interventions aiming to reduce the risk of mental health difficulties in marginalised groups*

Different types of tools appear to be effective in addressing individuals' psychological needs and may be well-suited for marginalised populations in pandemic time.

Bonardi et al. (2021) performed a review of nine randomised controlled trials (RCTs). Three were designed during the COVID-19 pandemic and included a racially and ethnically diverse sample. One trial tested the effects of a 4-week lay person-delivered intervention consisting of telephone calls to a group of homebound older adults in the USA receiving home meal services through the Meals on Wheels programme. The investigators trained university students in empathetic conversational skills (e.g., prioritising listening, eliciting conversation on topics of interest to participants) and each caller supported 6–9 participants. Calls were performed on 5 days during the first week and 2–5 days in the following 3 weeks and lasted less than 10 minutes. The study observed effects on participants' symptoms of anxiety, depression, overall mental health and loneliness. A second trial tested the efficacy of multifaceted videoconference-based 4-week group intervention tested in 12 countries and aiming to reduce levels of anxiety, depression, fear and loneliness among persons with rare autoimmune diseases, systemic sclerosis or scleroderma. The third study showed the effect of a self-guided online cognitive behavioural intervention tested in the general population of Sweden. Regarding these three interventions, COVID-19-specific anxiety and general anxiety symptoms were reduced by a standardised mean difference (SMD) of 0.31 (95% CI 0.03–0.58) to 0.74 (95% CI 0.58–0.90) compared to no intervention or a waitlist. Depressive symptoms were also reduced [SMDs between 0.31 (95% CI 0.05–0.70) and 0.56 (95% CI 0.22–0.55)].

Damiano et al. (2021) gathered 125 qualitative studies addressing preventive or interventional strategies to improve mental health and three RCTs adapted to the context of the COVID-19 pandemic. Most involved psychological/psychiatric interventions (12.8%), technological/media interventions (7.2%) (e-health or digital or telephone-based interventions), psychological/psychiatric interventions associated with technology/media and education (5.6%), self-care (exercise, eating habits, leisure time, sleep hygiene) and governmental programmes (5.6%). Using data from three RCTs with a total of 128 participants, the meta-analysis highlighted improvements in symptoms of anxiety, depression, sleep quality, hostility and somatisation [SMD = 0.87 (95% CI 0.33–1.41), $p < .001$, $I^2 = 69.2\%$]. The first trial was a 1-day group debriefing technique based on Asian philosophies and traditional Chinese medicine applied to persons who experience a chronic disease. The two other trials involved Internet-based interventions focusing on muscle and breathing relaxation for COVID-19 patients (30 minutes daily for 5 days; 2-week trial of daily 50-minute practices of breath relaxation techniques).

Rauschenberg et al. (2021) and Strudwick et al. (2021) found some evidence of effectiveness of digital general public interventions, which could be used in the context of the COVID-19 pandemic. According to these authors, e-health (electronic health) and m-health (mobile health) interventions decreased levels of symptoms of common mental disorders such as anxiety and depression during the COVID-19 pandemic. Additionally, these analyses appear to be cost-effective, although the number of studies is limited. Strudwick et al. also showed evidence that digital interventions are scalable and well-suited to the COVID-19 pandemic. These interventions had a positive impact on the risk of anxiety,

depression and PTSD. Notably, the authors included interventions directly designed for members of ethno-racial minority groups (3 articles) or groups experiencing socio-economic disadvantage (11 articles). The number of interventions tested in these marginalised groups is low, but this review showed the possibility of successful targeting. The key barriers to the implementation of such interventions are the difficulty to use technology, mistrust of technology or difficulty establishing a therapeutic alliance with healthcare providers due to technology-related challenges.

## Discussion

This literature review summarises data on the impact of the COVID-19 pandemic on the mental health of persons belonging to marginalised groups. Indeed, socio-economic disadvantage, as measured by a low educational level, low income or unemployment, is linked with the occurrence of mental health disorders such as anxiety, depression and acute stress. Similarly, migrants and persons belonging to ethno-racial minorities or experiencing homelessness are also at increased risk of psychological distress. Regarding actions that mitigate psychological distress, we found no reviews regarding specific interventions aiming to improve mental health during COVID-19 pandemic in marginalised groups. Yet, we found evidence of interventions that can be relevant for persons who are often excluded from standard healthcare. The systematic reviews we examined point to improvements in the prevention and care of mental health difficulties among members of marginalised groups which should be the focus of future research testing innovative community-based designs.

### Limitations

Our review has several limitations. First, we searched for relevant articles in PubMed and Google Scholar and may have bypassed some relevant publications. Nevertheless, Google Scholar covers multiple disciplines and fields, and it is unlikely that we missed important publications in the field. Second, the number of systematic reviews focused on the mental health of members of marginalised groups is limited, unlike healthcare workers, children, women, older people and people with pre-existing mental health disorders who have been widely studied (Sepúlveda-Loyola et al., 2020; Thibaut and Van Wijngaarden-Cremers, 2020; Vizheh et al., 2020; Jones et al., 2021; Hards et al., 2022). Third, there is a lack of variety among studies presented in the systematic reviews we examined. Most studies were cross-sectional, conducted online and come from industrialised countries such as China, USA or European countries. Therefore, there is need to additional high quality studies in the future. A notable systematic review performed by Jesline et al. (2021) focused on India and involved migrants, giving some information about the situation in a middle-income country. Fourth, marginalised groups were not equally represented in the scientific literature: we found few reviews dedicated to persons experiencing homelessness. Future research should take care to involve designs which make it possible to study marginalised groups which are often excluded from standard designs.

### Interpretation of study findings

Our findings imply a degree of universality in the relationship between membership in a marginalised group and mental health

difficulties in the context of the COVID-19 pandemic in Western countries. A relevant element, which was not discussed in the systematic reviews we analysed, is that marginalised groups may have experienced high levels of mental health difficulties prior to COVID-19 (Businelle et al., 2014; Silva et al., 2016). The COVID-19 outbreak may have exacerbated these pre-existing difficulties and highlighted the mental health needs in these high-risk groups. It is also relevant to point out that marginalised group may cumulate different forms of disadvantage (Wright et al., 2021). For example, a study conducted by Scarlett et al. (2021) shows that among persons experiencing homelessness, being unemployed was associated with the likelihood of depression alongside migrant status.

Interestingly, the COVID-19 pandemic led to unprecedented levels of resources for research on mental health. For example, the European *Preparedness of health systems to reduce mental health and psychosocial concerns resulting from the COVID-19 pandemic* (the RESPOND project) aims to identify vulnerable groups at highest risk of mental health problems and test an innovative stepped care intervention programme (Self-Help Plus and Problem-Management Plus developed by the World Health Organisation). The RESPOND project also examines the cost-effectiveness of this programme to identify effective strategies to improve health system preparedness in the event of a future pandemic (RESPOND Project, 2022). The RESPOND project aims to answer to several issues raised in this review: the need for screening tools which make it possible to identify groups at risk of suffering of psychological issues and the ways of addressing them, as indicated by Mendes-Santos et al. (2020); the necessity to test the use of digital and in-person psychosocial support interventions, as pointed out by Rauschenberg et al. (2021). The intervention being tested in the context of the RESPOND trial may be well-suited for populations who are marginalised and have difficulty accessing mental healthcare, and this aspect will be specifically examined. Finally, our literature review reveals the lack of data on marginalised populations in low- and middle-income countries, and the need for additional research in these settings (Table 2).

Nevertheless, despite existing gaps in knowledge, several recommendations can be made to healthcare professionals, on the basis on our findings as well as existing literature. First, given the increase in mental health problems and special vulnerability identified among persons who experience socio-economic disadvantage, homelessness or who are migrant, screening for symptoms of anxiety and depression in these groups should be disseminated as much as possible. Second, given difficulties in access to mental healthcare among members of socio-economically marginalised groups, there is need to develop innovative interventions to prevent but also address symptoms of anxiety and depression, are cost-effective and can be widely disseminated (Stewart and Appelbaum, 2020). In the aftermath of the COVID-19 pandemic, when the economic consequences start being evaluated, mental health professionals also need to be present on the public scene and in face of decision makers to indicate that even if the number of COVID-19 infections and death have decreased, the psychiatric consequences are far from over, particularly among marginalised groups, and will require budgetary and personnel commitments to be addressed (McDaid, 2021).

## Conclusion

The mental health of marginalised groups was severely affected during the COVID-19 pandemic. Symptoms of mental health

**Table 2.** Interventions aiming to address mental health needs in the context of COVID-19 suitable for marginalised groups

| Authors (year) | Journal | Settings/purpose of the review | Characteristics of the study/study population | Type of intervention | Outcomes/ recommendations/ conclusions |
|---|---|---|---|---|---|
| *Interventions and recommendations* | | | | | |
| Strudwick et al. (2021) | *Journal of Medical Internet Research* | Digital interventions that could be used to support the mental health | Description of different digital interventions suitable in the context of the COVID-19 pandemic. | Mobile app (31) and Internet-based resources (114) | A variety of digital interventions were identified. The authors suggested the need to develop targeted intervention and evaluate their cost-effectiveness. The authors also point out that few apps addressed equity-related considerations |
| Bonardi et al. (2021) | *The Canadian Journal of Psychiatry* | Randomised controlled trial of an Internet based programme | Nine eligible trials: 3 interventions designed specifically in the context of mental health challenges during the COVID-19 pandemic, 6 trials of standard interventions (e.g., individual or group therapy, expressive writing, mindfulness recordings) minimally adapted for COVID-19 | Internet-based mental health programmes for the general population and lay- or peer-delivered interventions for marginalised groups may be effective and scalable in the context of COVID-19 | A self-guided Internet intervention reduced both anxiety and depression symptoms, consistent with a growing body of evidence that Internet-based psychological interventions may be effective as first-line strategies for many people scalable in the context of the COVID-19 crisis |
| Rauschenberg et al. (2021) | *Journal of Medical Internet Research* | Gathering of a theoretical and empirical base, user perspective, safety, effectiveness and cost-effectiveness of digital interventions related to public mental health provisions (i.e. mental health promotion, prevention and treatment of mental disorders) that may help to reduce the consequences of the COVID-19 pandemic | Analysis of systematic reviews and meta-analyses that investigated digital tools for mental health promotion, prevention and treatment of mental health conditions and determinants likely affected by the COVID-19 pandemic | Telemedicine, e-health, m-health | The scientific literature shows that digital interventions can be effective in the context of the COVID-19 pandemic to reduce symptoms of depression and anxiety. However, there is a lack of cost-effectiveness studies |
| Damiano et al. (2021) | *Brazilian Journal Psychiatry* | Review of the most common mental health strategies aimed at alleviating and/or preventing mental health problems in the context of the COVID-19 pandemic | Preventive or interventional strategies for mental health symptoms in individuals during coronavirus pandemics | One-day group debriefing technique, Strength-Focused and Meaning Oriented Approach for Resilience and Transformation (SMART), based on Asian philosophies and traditional Chinese medicine applied for people with chronic diseases 1 month after the SARS pandemic (1RCT); progressive muscle relaxations (2 RCT) | The present review found few clinical trials assessing the effectiveness of interventions to improve the mental health of individuals during the COVID-19 pandemics. Qualitative articles identified highlight the potential effectiveness of digital interventions |
| Mendes-Santos et al. (2020) | *Frontiers* | This review presents an overview of initiatives developed to address the challenges faced by the mental healthcare system, and discusses how the timely | Description of innovative psychosocial intervention strategies in Portugal | Recommendations in these domains: Internet research, screening and tracking tools, tele-counselling and psychotherapy, Internet interventions, | The authors described an initiative in Portugal and recommended ways to improve the mental healthcare of Portuguese people during the COVID-19 pandemic |

**Table 2.** (*Continued*)

| Authors (year) | Journal | Settings/purpose of the review | Characteristics of the study/study population | Type of intervention | Outcomes/ recommendations/ conclusions |
|---|---|---|---|---|---|
| | | implementation of a comprehensive digital mental health strategy, coupling research, education, implementation and quality assessment initiatives might buffer the impact of the COVID-19 pandemic | | comprehensive e-Learning and e-supervision initiatives | |

disorders reported and studied predominantly included anxiety, depression and stress. Public authorities should be aware of the role of socio-economic disadvantage, being a migrant, being a minority and/or being homeless with regard to poor mental health, especially during a pandemic such as COVID-19. This should alert healthcare providers and policy makers as to the need regarding specific support required by these groups. Mental health interventions should also better target these marginalised groups in order to mitigate psychological distress. Indeed, interventions mitigating mental health distress in times of crisis exist and should be developed and rendered appropriate to marginalised populations. Although marginalised groups were disproportionally impacted, few studies and interventions specifically target this group. More targeted studies and interventions are needed to reduce social inequalities with regard to mental health difficulties in the aftermath of the COVID-19 pandemic and to prepare possible future outbreaks.

**Open peer review.** To view the open peer review materials for this article, please visit https://doi.org/10.1017/gmh.2022.56.

**Data availability statement.** All data used in this publication are available upon request.

**Author contributions.** M.M. had the original idea for the study and discussed it at length with all authors. C.C. conducted the literature review, synthesised the studies found and drafted an initial version of the manuscript. M.M. finalised the manuscript, which was read, revised and approved by all authors.

**Financial support.** This study was made possible owing to funding by the EU Horizon 2020 H2020-SC1-PHE-CORONAVIRUS-2020-2 programme (RESPOND – Improving the Preparedness of Health Systems to Reduce Mental Health and Psychosocial Concerns resulting from the COVID-19 Pandemic project, Grant Agreement 101016127). P.S. benefitted from support from the Paris Institute for Advanced Study (France).

**Competing interests.** The authors declare no competing interests exist.

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
