## [Reviewer Report]

*Comments to Author*: General comments

The main aim of the research paper is to explore the literature in order to gain a deeper understanding of the impacts of COVID-19 on mental health of persons belonging to marginalized groups, as well as mapping interventions aimed to reduce such impacts. The authors successfully underline the aim of the paper by using adequate background literature to support their case. The study is carried out via scoping review, authors have chosen appropriate databases (Google Scholar and PubMed), although adding one more academic database such as Ovid could have benefitted the scope of the evidence found. The authors conclude that global literature has underlined the negative impacts of COVID-19 on mental health of marginalized populations. Authors also present interventions which may be suitable for adaptation for marginalized populations. Finally, the authors appropriately identify key limitations of the presented literature. Overall this is an excellent review in which authors have successfully and succinctly presented the available literature. Furthermore, the paper adds to a body of research on a population which has been overlooked, as the author points out. Some minor revisions below, could allow the authors to clarify the information given within their paper.

Detailed comments by section

Title and keywords

The title adequately describes the research; however, it describes “psychological distress” when in fact the rest of the paper uses terms such as “psychological distress and psychiatric disorders” or “mental health impacts”. I would suggest using the broad term “mental health” in the title in order to cover all outcomes that are described within the paper.

Introduction

The introduction provides good background on some of the concepts being studied, this makes the operationalization of each of these topics within the current study clear.

However, the introduction lacks background literature on interventions for the impacts of COVID-19. Since mapping interventions is one of the aims it would be adequate to provide some background on such interventions within the introduction.

Methods

The methods section provided ample definitions for all searched components. Perhaps the authors could add a rationale to better explain why mental health outcomes were defined as “anxiety, depression, stress, and psychological distress”.

As already mentioned in the general comments, adequate databases were chosen. Nevertheless, the addition of one more academic database would allow for a larger scope in the search results, this is however not a major issue as Google Scholar and PubMed are large databases which allow for an already broad scope of search. Authors should not this as a limitation to their study.

The search strategy is well presented; however, the reader would benefit from having access perhaps within the appendices to the full search strategy as operationalized for the respective databases.

On page 4 one of the search terms is misspelled as “SRAS-CoV-2”.

Results and discussion

The result and discussion sections provide a good summary of the findings. The author links the results to the existing literature and provides further detail and reach to the findings.

On page 14 within the conclusion section there is a misspelling: “being a minority and/or being homelessness with regard to poor mental health”.

---

## [Reviewer Report]

*Comments to Author*: This is an interesting scoping review on the impact of the COVID-19 pandemic on marginalized groups, which have been overlooked during the world health crisis.

Although the paper is interesting and well-written, I have some concerns which need to be addressed:

1. authors should provide a rigorous definition of "marginalized group",also providing some relevant quotation. Authors should explain the reason for not including studies of sexual minorities (e.g., the LGBTQI+ community is often considered a marginalized group), and also should mention that some papers on such minorities already exist.

2. In the Introduction, authors should describe the reasons for considering the pandemic a new form of trauma for mental health and how it can be managed (i.e., Adhanom Ghebreyesus T. Addressing mental health needs: an integral part of COVID-19 response. World Psychiatry. 2020 Jun;19(2):129-130. doi: 10.1002/wps.20768. PMID: 32394569; PMCID: PMC7214944; Bridgland VME, Moeck EK, Green DM, Swain TL, Nayda DM, Matson LA, Hutchison NP, Takarangi MKT. Why the COVID-19 pandemic is a traumatic stressor. PLoS One. 2021 Jan 11;16(1):e0240146. doi: 10.1371/journal.pone.0240146. PMID: 33428630; PMCID: PMC7799777)

3. A PRISMA flowchart should be included in order to describe the selection process of the included papers.

4. Have you evaluated the quality of the included systematic reviews using the AMSTAR tool? If not, you should consider to use it or to provide a rationale for not using.

5. Practical implications for health professionals should be discussed, also taking into consideration some recent statements issued by international scientific society such as the World Psychiatric Association and the European Psychiatric Association (i.e., Stewart DE, Appelbaum PS. COVID-19 and psychiatrists' responsibilities: a WPA position paper. World Psychiatry. 2020 Oct;19(3):406-407. doi: 10.1002/wps.20803. PMID: 32931089; PMCID: PMC7491607; McDaid D. Viewpoint: Investing in strategies to support mental health recovery from the COVID-19 pandemic. Eur Psychiatry. 2021 Apr 26;64(1):e32. doi: 10.1192/j.eurpsy.2021.28. PMID: 33971992; PMCID: PMC8134893.

---

## [Reviewer Report]

*Comments to Author*: Additional two requests for authors:

1. Full search strategy shall be provided otherwise it is difficult to assess its accuracy. 

2. Established terminology should be adhered to - the term "umbrella review" is preferred over "scoping review of systematic reviews".

---

## [Reviewer Report]

Background: The CovidCOVID-19 pandemic has had negative consequences on the mental health of the population, which has been documented. Marginalized groups, that are at risk of poor mental health overall, have been particularly impacted. 

Objective: The purpose of this review is to describe the mental health impact of the CovidCOVID-19 pandemic on marginalized group (that is persons who are socio-economically disadvantaged, migrants and members of ethno-racial minorities, experience homelessness) and identified interventions which could be well-suited to prevent and address mental health difficulties. 

Methods: We conducted a scoping review of systematic reviews on mental health difficulties since the beginning of the COVID-19 epidemic and appropriate interventions among marginalized groups published from January 1st 2020 up to May 2, 2022, using Google Scholar and PubMed (MEDLINE). 

Results: Among 792 studies on mental health difficulties among members of marginalized groups identified by keywords, 17 studies met our eligibility criteria. 12 systematic reviews examining mental health difficulties in one or several marginalized groups during the COVID-19 pandemic and 5 systematic reviews on interventions that can mitigate the mental health impact of the COVID-19 pandemic were retained in our scoping review. 

Conclusion: The mental health of marginalized groups was severely affected during the CovidCOVID-19 pandemic. Most frequently reported mental health difficulties included symptoms of anxiety and depression. Additionally, there are interventions that appear effective and well-suited for marginalized populations, which should be disseminated on a large scale to mitigate the psychiatric burden in these groups and at the population level.

---

## [Reviewer Report]

*Comments to Author*: thank you for this revised version of the paper and for addressing my comments

---

## [Reviewer Report]

*Comments to Author*: Recommendation: "scoping review" with "umbrella review" or with any other established term for a review of reviews.